# Variations in Intraocular Pressure Among Athletes Across Different Sports Disciplines

**DOI:** 10.3390/jcm14093211

**Published:** 2025-05-06

**Authors:** Feliciana Menna, Laura De Luca, Stefano Lupo, Alessandro Meduri, Enzo Maria Vingolo

**Affiliations:** 1Department of Medical-Surgical Sciences and Biotechnologies, U.O.C. Ophthalmology, Sapienza University of Rome, Via Firenze 1, 04019 Terracina, Italy; stelup@hotmail.com (S.L.); enzomaria.vingolo@uniroma1.it (E.M.V.); 2Department of Biomedical and Dental Science and of Morphological and Functional Images, University of Messina, 98122 Messina, Italy; laura.deluca21@gmail.com (L.D.L.); ameduri@unime.it (A.M.)

**Keywords:** intraocular pressure (IOP), athletes, sports disciplines, glaucoma

## Abstract

**Objectives**: Elevated intraocular pressure (IOP) is a well-known risk factor for glaucoma. This study investigated the impact of two distinct types of physical activity—endurance (marathon running) and strength (weightlifting)—on IOP variations. **Methods**: Forty healthy male athletes (20 marathon runners, 20 weightlifters) aged 18–35 years were recruited and monitored over three months. IOP was measured using Goldmann and Icare IC200 tonometers before and after 1 h training sessions. **Results**: The results showed a significant increase in IOP after training among weightlifters (mean post-training IOP: 19.3 mmHg), in contrast to stable or slightly reduced values in marathon runners (mean post-training IOP: 15.1 mmHg). **Conclusions**: These findings suggest the need for regular ophthalmologic monitoring in strength athletes. Future studies should examine the long-term impact of sport-specific IOP fluctuations on ocular health and glaucoma risk.

## 1. Introduction

Intraocular pressure (IOP) is a key physiological parameter essential for ocular homeostasis. Persistent IOP elevation remains the most significant modifiable risk factor for glaucomatous optic neuropathy, a leading cause of irreversible blindness worldwide [1]. Accordingly, identifying factors that influence IOP—particularly those related to lifestyle and physical activity—is critical in both glaucoma prevention and management.

Physical exercise is widely endorsed for its systemic benefits, including enhanced cardiovascular function, metabolic regulation, and reduced systemic inflammation and stress [2]. However, its ocular effects remain less clearly understood and have become an area of growing clinical interest. While early studies suggested a transient IOP reduction following exercise in glaucoma patients [3], later research has highlighted a more nuanced relationship. The IOP response appears to depend on variables such as exercise type, intensity, duration, and individual physiology [4].

Recent investigations have focused on differentiating the ocular effects of various exercise modalities [5,6]. Endurance training—characterized by aerobic activities like long-distance running—is generally associated with IOP reductions, possibly due to improved ocular perfusion, decreased sympathetic activity, and lower plasma osmolarity [7]. In contrast, anaerobic or isometric activities, such as weightlifting, have been linked to transient IOP elevations [8,9], likely driven by Valsalva-like maneuvers, increased intrathoracic and episcleral venous pressures, and the transient obstruction of aqueous humor outflow [7,10,11].

This physiological divergence has important implications for athletes, particularly those engaged in high-resistance training. For instance, Vaghefi et al. (2022) reported IOP spikes exceeding 25 mmHg during leg press exercises, raising concerns about potential cumulative optic nerve stress in susceptible individuals [10]. Conversely, Najmanova et al. (2021) found that endurance training may stabilize IOP and offer potential protective benefits [12].

Despite emerging evidence, comparative studies examining IOP responses across distinct athletic disciplines remain scarce, often limited by heterogeneous methodologies or a lack of sport-specific analysis. To address this gap, the present study systematically evaluates the effects of two contrasting training regimens—endurance (marathon running) and strength (weightlifting)—on IOP in healthy young male athletes. IOP was measured using both Goldmann applanation and rebound tonometry (Icare IC200) before and after standardized 1 h training sessions over a three-month period.

The primary objectives of this study are (1) to determine whether these distinct forms of exercise elicit differential IOP responses and (2) to explore the broader implications for long-term ocular health, including glaucoma risk. With increasing global participation in competitive and recreational sports, these findings aim to inform clinicians, trainers, and athletes about the importance of incorporating sport-specific ocular monitoring into routine health assessments.

## 2. Materials and Methods

### 2.1. Study Design and Participants

This study was designed as a prospective, observational investigation aiming to assess intraocular pressure (IOP) changes associated with two distinct athletic disciplines—endurance running and weightlifting. The research was conducted at the Department of Sports Medicine, University of Campania ‘Luigi Vanvitelli’ (Napoli, Italy), from September 2019 to July 2020 and at the Department of Ophthalmology, Alfredo Fiorini Hospital of Terracina (Latina, Italy), from September 2023 to May 2024. Ethical approval was obtained from the institutional review board, and written informed consent was secured from all participants prior to enrollment.

A total of 40 healthy male athletes were recruited through local sports organizations and fitness centers during routine sports fitness assessments. Although the sample size of 20 athletes per group provides preliminary insight into IOP responses to different exercise modalities, the limited statistical power restricts the generalizability and robustness of regression-based inference. No formal power analysis was conducted a priori, which is a limitation that should be addressed in future studies to ensure adequate effect size estimation and Type II error control. However, a post hoc power analysis was performed using GPower 3.1 software, which indicated that with a sample size of 20 per group, the study achieved 76% power to detect a medium effect size (*d* = 0.7) at an alpha level of 0.05 for paired comparisons. This suggests moderate statistical power, though underpowered for detecting smaller effects or conducting reliable multivariable regression analyses.

Participants were required to be between 18 and 35 years of age and actively engaged in either marathon training (*n* = 20) or weightlifting (*n* = 20), with each group representing the respective endurance and strength training modalities. All enrolled individuals were in their first year of competitive practice to ensure a relatively uniform training background and minimize the potential confounding effects of long-term adaptation.

To ensure the internal validity of the study, strict inclusion and exclusion criteria were applied. Inclusion criteria mandated male gender, an age range of 18–35 years, the absence of previous competitive experience in the selected sport, and no known history of ocular disease or previous ocular surgery. Additionally, participants were required to have normal baseline IOP readings (defined as ≤19 mmHg) at the time of enrollment, confirmed through initial tonometric assessment.

Exclusion criteria were established to eliminate potential confounders. Athletes with significant refractive errors (greater than ±3 diopters of myopia or hyperopia), those using systemic or topical medications known to influence IOP (such as corticosteroids or beta-blockers), and individuals with systemic conditions like diabetes mellitus or hypertension were excluded. These conditions are known to independently impact IOP or optic nerve health and could skew the study’s findings. By carefully selecting a homogenous and healthy study population, the study aimed to isolate the effects of exercise type on IOP fluctuations with minimal bias.

### 2.2. Clinical Assessment

All participants underwent a standardized battery of clinical assessments to establish baseline physiological and ocular health parameters. These evaluations were conducted by trained personnel under the supervision of the study’s principal investigator (F.M.).

Cardiovascular function was assessed using a 12-lead electrocardiogram (ECG), performed in accordance with the European Society of Cardiology guidelines. Bioimpedance analysis (BIA) was utilized to determine body composition, including body fat percentage and lean muscle mass, using a validated multi-frequency segmental device (Tanita MC-780U, Tanita Corporation, Tokyo, Japan). Pulmonary function testing was also performed, with specific attention to peak expiratory flow (PEF), using a portable spirometer calibrated for daily use. These metrics served to document the athletes’ baseline fitness and overall health status.

A comprehensive ophthalmologic examination was conducted for all participants to establish ocular baseline values and exclude pre-existing pathology. Best-corrected visual acuity (BCVA) was measured using a standardized Snellen chart under photopic conditions. Fundus examination was carried out using direct ophthalmoscopy (Heine Beta 200, HEINE Optotechnik, Gilching, Germany), focusing on the optic nerve head and macula to exclude glaucomatous changes or other macular pathology.

Intraocular pressure was assessed using the following two methods to ensure reliability and reproducibility: the Goldmann applanation tonometer (Haag-Streit, Bern, Switzerland), considered the gold standard in clinical practice, and the Icare IC200 rebound tonometer (ICare Finland Oy, Helsinki, Finland), which allows for quick, minimally invasive measurements. Each participant underwent IOP measurements with both devices in a single session, spaced five minutes apart to minimize any inter-device bias. All measurements were conducted by a single ophthalmologist (F.M.) to minimize operator-dependent variability. For each eye, four IOP readings were taken and averaged to obtain a reliable measurement. The mean value from both eyes was used in the final analysis.

These comprehensive assessments ensured a robust clinical foundation for evaluating the relationship between sport-specific physical activity and IOP variations.

### 2.3. IOP Measurement Protocol

To ensure consistency and reduce measurement variability, intraocular pressure (IOP) assessments were carried out under controlled conditions and by the same experienced ophthalmologist (F.M.) throughout the study. All baseline measurements took place in the hospital’s ophthalmology unit, and pre- and post-training measurements were conducted on-site at the sports center, where a slit-lamp equipped with a Goldmann tonometer was temporarily installed.

Participants were instructed to avoid caffeine intake, alcohol consumption, and vigorous physical activity for at least 12 h prior to testing. On each measurement day, athletes rested in a seated position for a minimum of 10 min before the first IOP reading to allow for hemodynamic stabilization and to avoid artificially elevated values due to recent exertion or positional changes.

IOP was measured at the following three timepoints: during the baseline visit before the initiation of any physical activity; immediately before a scheduled 1 h training session, either endurance running or resistance weightlifting; and within five minutes following the completion of the training session to capture acute post-exercise changes.

IOP was measured immediately after training to capture acute fluctuations. However, the lack of medium- and long-term follow-up prevents the assessment of IOP recovery dynamics, which could reveal cumulative ocular stress in athletes with frequent training cycles. Future protocols should include post-exercise recovery measurements at 15, 30, and 60 min to better characterize IOP trends over time.

Two different devices were used during each measurement session to ensure the reliability and cross-validation of the readings. The Goldmann applanation tonometer, mounted on a slit-lamp biomicroscope, was used after instilling topical anesthesia and fluorescein dye. Use of the Icare IC200 rebound tonometer, a handheld and minimally invasive device, allowed for rapid assessment without the need for anesthetic. To minimize bias, a five-minute interval was maintained between the use of the two devices, and the order of device usage was alternated across subjects.

For each eye, four consecutive readings were taken using both devices. These values were averaged to obtain a representative IOP for each eye, and the mean of both eyes was used as the subject’s final IOP value at each timepoint. This methodological approach ensured the high reliability of data, while controlling for variability due to environmental, procedural, or equipment-related factors.

### 2.4. Statistical Analysis

All statistical analyses were performed using IBM SPSS Statistics software, version 25.0 (IBM Corp., Armonk, NY, USA). Continuous variables were expressed as means with standard deviations, and categorical variables were presented as frequencies and percentages. The distribution of continuous data was evaluated using the Shapiro–Wilk test to assess normality.

As the variables demonstrated normal distribution, parametric tests were applied throughout. Differences in baseline demographic and physiological characteristics between the marathon runner and weightlifter groups were assessed using independent samples Student’s *t*-tests. Categorical comparisons were evaluated using Pearson’s chi-square test where applicable.

Intra-group differences in IOP values before and after training were analyzed using paired *t*-tests. Inter-group comparisons of post-exercise IOP levels and changes from baseline were analyzed with independent *t*-tests. Statistical significance was defined as a *p*-value less than 0.05.

In addition to primary comparisons, Pearson correlation coefficients were calculated to explore the relationships between changes in IOP and other variables, such as body mass index (BMI), peak expiratory flow (PEF), and subjective exertion scores based on the Borg scale. Exploratory linear regression models were also generated to investigate the potential predictors of post-exercise IOP elevation, particularly in the weightlifting group; although, these analyses were considered secondary and exploratory in nature.

## 3. Results

For all athletes participating in the study, during the period between September 2019 and July 2020 and between September 2023 and May 2024, the electrocardiogram (ECG) revealed no significant alterations, and the heart rate ranged between 58 and 70 bpm, with a standard deviation of ±5 bpm. The average body mass index (BMI) of the marathon runners was 22.5 kg/m^2^ (±1.2 kg/m^2^), indicating a lean body composition typical of endurance athletes. In contrast, the weightlifters had a mean BMI of 27.3 kg/m^2^ (±1.5 kg/m^2^), suggesting greater muscle mass. The mean peak expiratory flow (PEF) values for the marathon runners were 600 L/min (±30 L/min), indicating good respiratory capacity and lung function. Weightlifters recorded an average PEF of 550 L/min (±25 L/min) (Table 1).

Regarding IOP, all athletes underwent IOP assessment of both eyes by the same operator. The IOP was recorded as the average of four consecutive readings with the eye in a primary position; the right eye was always measured first. The average between the tonometric values of the two eyes was then obtained.

At the first visit, in the 20 selected marathon runners, the mean IOP was 15.2 mmHg (±1.0 mmHg). In the 20 weightlifters, the mean IOP at the first visit was 16 mmHg (±0.9 mmHg), slightly higher than in the marathon runners.

Before training, the mean IOP was 15.4 mmHg (±0.8 mmHg) in the marathon runners, showing only a small variation. Under the same conditions, the mean IOP for weightlifters was 16.2 mmHg (±1.0 mmHg), showing substantial stability compared to the first visit.

After 1 h of training, the mean IOP of the marathon runners stabilized at 15.1 mmHg (±0.9 mmHg), while the mean IOP for the weightlifters was 19.3 mmHg (±2.1 mmHg), indicating an increase in IOP compared to the values before training (Table 2). For 30% of the athletes, this increase was significant by more than 4 mmHg (*p*-value = 0.01) (Figure 1).

The regression analysis in the weightlifting group revealed that the peak exertion level (maximum workload) was a significant predictor of post-exercise IOP (R^2^= 0.32, *p* = 0.04), indicating a modest but statistically significant relationship (Figure 2). No significant predictors were identified in the endurance group.

The Cohen’s d effect size for the mean IOP increase in weightlifters from pre- to post-training was calculated at 1.88, indicating a large effect. In contrast, the change in marathon runners was negligible, with a Cohen’s *d* of −0.33, suggesting a small and clinically insignificant decrease.

The group of marathon runners, therefore, showed a negative correlation between training duration and IOP, suggesting that an increase in training volume could be associated with a lowering of IOP. In the group of weightlifters, on the other hand, a positive correlation was noted between maximum workload and IOP, indicating that greater exertion may contribute to increased IOP values.

The differences in tonometric values between the two groups can be explained by several factors. During weightlifting, athletes may experience an increase in intraocular pressure due to intra-abdominal hyperpressure, which may be reflected in IOP [7]. Marathon runners, who tend to maintain regular, low-effort breathing, may benefit from better IOP control [8]. Training outdoors (for marathon runners) versus indoors (for weightlifters) could further influence results [9].

## 4. Discussion

The present study highlights that IOP responses differ significantly depending on the type and intensity of athletic activity. Endurance athletes, such as marathon runners, tended to maintain stable or slightly reduced IOP following exercise, while weightlifters showed notable post-training elevations. These results are consistent with previous findings that physical activity influences IOP but reinforce that not all exercise types confer a protective effect [13,14].

We adopted a ≥4 mmHg IOP increase threshold based on existing literature citing transient, exercise-induced pressure spikes with potential clinical importance, though standardized criteria remain lacking [15]. Future studies should align such thresholds with clinical guidelines or long-term risk models for glaucoma progression.

The more stable IOP observed in marathon runners may reflect the physiological benefits of aerobic activity—namely, improved cardiovascular efficiency, vasodilation, and consistent breathing patterns—all of which may promote ocular perfusion and reduce pressure variability [16]. Additionally, the outdoor nature of endurance training may contribute via environmental factors, such as light exposure and ambient variability [17]. Although marathon running is predominantly aerobic, certain phases—such as sprinting at the finish line, navigating inclines, or during competitive pacing—can activate anaerobic energy systems. This intermittent reliance on anaerobic metabolism has been observed even in endurance athletes, particularly in middle-distance running, where lactate accumulation and oxygen debt are more prominent [18]. These physiological transitions may influence intraocular pressure (IOP) differently than steady-state aerobic exertion. Future research should investigate IOP responses in athletes performing high-intensity interval training or variable-intensity running to better characterize the ocular impact of mixed aerobic–anaerobic workloads.

Conversely, IOP elevation in weightlifters may be influenced by the intense nature of strength-based exercise. Valsalva-like maneuvers, common during resistance training, can elevate intrathoracic and intra-abdominal pressure [19], which in turn, may raise episcleral venous pressure and temporarily hinder aqueous outflow, contributing to IOP spikes [20]. While these mechanisms are plausible, their direct contribution warrants further investigation. Our findings are in line with prior reports of transient IOP increases during high-resistance exercises, such as leg presses [10,11].

Psychological and hormonal responses to intense training may also influence IOP. Physical stress activates the hypothalamic–pituitary–adrenal (HPA) axis, increasing cortisol levels, which may affect aqueous humor dynamics and contribute to transient IOP elevation [15,16,17,18]. In support of this, Vera et al. showed that IOP rises with anxiety during high-pressure athletic scenarios [21].

The long-term implications of transient IOP fluctuations are of clinical interest. Even short-term spikes may cumulatively strain the optic nerve in glaucoma-susceptible individuals. IOP variability has been associated with disease progression [22], and repetitive pressure elevations—such as those seen in wind instrument players—have been linked to visual field deterioration [10].

Muscular exertion itself may contribute to IOP elevation. Haruna et al. noted that Valsalva-induced IOP spikes were paralleled by increased electromyographic activity [23]. The role of extraocular muscle tension, including effects from lid squeezing, further illustrates the complexity of exercise-related IOP changes [12,24].

Notably, IOP responses may vary by individual due to genetics, baseline fitness, and pre-existing ocular conditions [19,20]. As Najmanova et al. observed, post-exercise IOP changes are modulated by initial heart rate and resting pressure [21]. Even within athletic cohorts, differential risk profiles likely exist. Sustained isometric effort, common in weightlifting, has been shown to raise IOP in concert with systemic blood pressure [25,26,27]. In contrast, endurance training appears to support vascular flexibility and IOP stability [14,28].

Emerging evidence also suggests that exercise may exert neuroprotective effects, including the upregulation of brain-derived neurotrophic factor (BDNF) and improved mitochondrial function, potentially guarding against retinal ganglion cell damage in glaucoma [29,30]. While our study did not evaluate these parameters, their relevance is promising.

Several physiological mechanisms, including intra-abdominal pressure, vascular resistance, and respiratory patterns, were considered in interpreting exercise-related IOP fluctuations. While prior literature has implicated endocrine and neuroprotective pathways, such as cortisol modulation and BDNF in ocular responses, these were not assessed directly. Future studies incorporating biomarker analysis will be critical for elucidating these links.

## 5. Conclusions

This study highlights the divergent effects of endurance and strength training on IOP. Marathon running appears to exert a stabilizing or mildly beneficial influence on IOP; whereas, weightlifting is associated with acute post-exercise elevations. These observations carry important implications for individuals at elevated risk of glaucoma, as repeated transient IOP spikes may contribute to cumulative optic nerve stress, particularly in those with predisposing factors. Accordingly, regular ophthalmologic evaluations—including IOP monitoring—may be advisable for strength-trained athletes. The integration of ocular assessments into routine sports medicine evaluations could facilitate early detection and prevention strategies, especially for individuals engaged in high-resistance training.

Several limitations of this study warrant consideration. The sample was relatively small and homogeneous, comprising exclusively young male athletes in their initial year of competitive training. Although this design minimized variability related to training experience and hormonal fluctuations [31], it limits the generalizability of the findings. Future research should aim to include broader and more diverse populations, encompassing female athletes, older individuals, and varying levels of training exposure. Additionally, the cross-sectional nature of the study and limited observation period restrict the ability to infer long-term ocular outcomes. Longitudinal studies incorporating extended follow-up, larger cohorts, and the evaluation of systemic biomarkers, hormonal profiles, and genetic predispositions are needed to further elucidate the mechanisms driving IOP variability in athletic settings. Such efforts will be essential in informing evidence-based exercise guidelines that safeguard both systemic and ocular health.

## Figures and Tables

**Figure 1 jcm-14-03211-f001:**
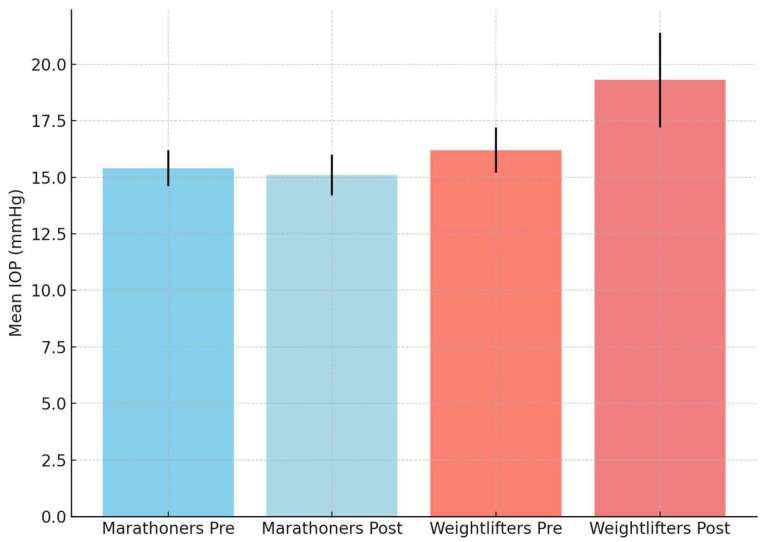
Mean intraocular pressure (IOP) values before and after training in marathon runners and weightlifters. Error bars represent standard deviation.

**Figure 2 jcm-14-03211-f002:**
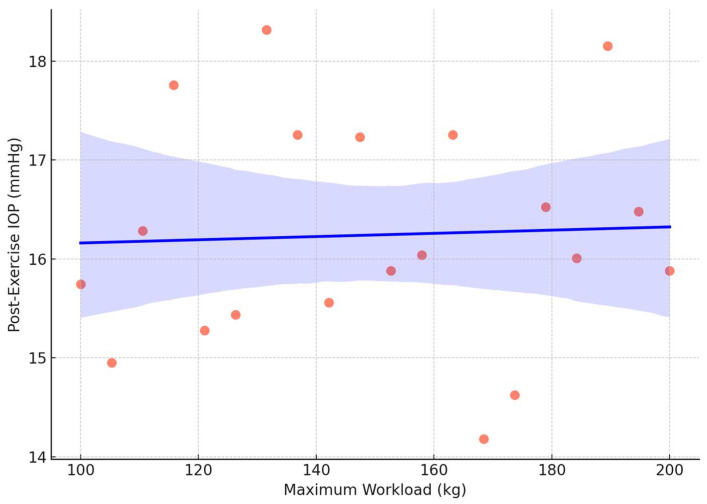
Scatter plot illustrating the relationship between maximum workload and post-exercise intraocular pressure (IOP) in weightlifters. Each orange dot represents an individual participant’s data point. The blue line depicts the linear regression trend, while the surrounding purple shaded area indicates the 95% confidence interval. A significant positive correlation was observed (R^2^ = 0.32, *p =* 0.04), suggesting that higher exertion levels are associated with increased post-training IOP.

**Table 1 jcm-14-03211-t001:** Clinical characteristics of study patients expressed as mean ± standard deviation (SD).

	Marathoners	Weightlifters
Age, mean ± SD, years	27.15 ± 4.7	28.8 ± 3.5
BMI ***^1^**, mean ± SD, kg/m^2^	22.5 ± 1.2	27.3 ± 1.5
PEF ***^2^**, mean ± SD, L/min	600 ± 30	550 ± 25

***^1^** BMI = body mass index; ***^2^** PEF = peak expiratory flow.

**Table 2 jcm-14-03211-t002:** Mean IOP *^1^ of study patients on first visit, before training, and after training.

	Marathoners	Weightlifters
IOP on first visit, mean ± SD, mmHg	15.2 ± 1.0	16 ± 0.9
IOP before training, mean ± SD, mmHg	15.4 ± 0.8	16.2 ± 1.0
IOP after training, mean ± SD, mmHg	15.1 ± 0.9	19.3 ± 2.1

***^1^** IOP = intraocular pressure, SD = standard deviation.

## Data Availability

The original contributions presented in this study are included in the article. Further inquiries can be directed to the corresponding author.

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
