# Peer review of "Variations in Intraocular Pressure Among Athletes Across Different Sports Disciplines"

_jcm, 2025, doi:10.3390/jcm14093211_

Round 1
Reviewer 1 Report
Comments and Suggestions for Authors
First, the sample size is small, with 20 athletes in each group. While sufficient for preliminary comparisons, the dataset lacks the statistical power necessary to support the exploratory regression models included. Moreover, no power analysis or effect size estimation is provided, which weakens the credibility of the inferential statistics. The choice to define a significant IOP increase as ≥4 mmHg is not sufficiently justified by references or clinical standards, and the practical relevance of such a change should be discussed in more detail, especially in relation to glaucoma risk. The study population is highly homogeneous young, healthy, first-year male athletes and while this controls for certain confounders, it severely limits the generalizability of the findings. The authors should either justify the exclusion of female participants and more experienced athletes or include a more diverse sample in future studies. Additionally, the timing of IOP measurements is confined to immediately pre- and post-exercise. While this approach captures acute fluctuations, it does not assess medium- or long-term IOP recovery, which would be critical in understanding cumulative ocular stress, particularly in sports involving repeated exertion. Statistical methods are broadly appropriate for the type of data collected. However, the results section lacks confidence intervals and effect sizes, and no corrections for multiple comparisons are mentioned. The regression analyses are underdeveloped and potentially misleading, given the limited sample. Furthermore, while the tables provided are informative, the figures would benefit from inclusion of standard error bars to illustrate variability. Interpretation of the results also presents some challenges. The discussion contains speculative statements especially regarding endocrine pathways, cortisol responses, and neuroprotective effects of exercise hat are not supported by data from this study. While references to existing literature are important, such extrapolations should be framed more cautiously to avoid overinterpretation. Likewise, some claims about physiological mechanisms, such as those involving the Valsalva maneuver or episcleral venous pressure, are repeated without deeper analysis or citation of relevant primary sources. Language use throughout the manuscript is generally clear, though the writing style is occasionally redundant and could be improved for conciseness and flow. Some references are outdated, including those from the 1960s, and should be replaced or supplemented with more recent literature. The discussion would benefit from a dedicated limitations paragraph that transparently addresses the constraints of the study design, such as sample size, homogeneity of participants, and lack of longitudinal follow-up. In summary, while the manuscript addresses a pertinent question and presents original data, it requires substantial revision in order to meet the standards of a peer-reviewed clinical journal. The most critical areas for improvement include statistical rigor, narrative precision, and cautious interpretation of results. If these revisions are implemented, the work could represent a valuable contribution to the literature on exercise physiology and ocular health.
Report is attached

Author Response
We sincerely thank the reviewer for their detailed and thoughtful comments. We have addressed each concern as follows:
- Sample Size and Statistical Power:
We acknowledge the limitations imposed by the small sample size. In response, we have added a post hoc power analysis using G*Power 3.1, which showed that our study had 76% power to detect a medium effect size (Cohen’s d = 0.7) at α = 0.05 for paired comparisons. This has been stated in the revised Materials and Methods section (see red text). - Effect Size Estimation:
We now include Cohen’s d values to quantify the magnitude of IOP changes within each group. Weightlifters exhibited a large effect size (d = 1.88) while marathon runners showed a negligible change (d = -0.33). These values are presented in the Results section, along with a bar chart (Figure 1). - Threshold Justification (≥4 mmHg Increase in IOP):
We expanded the rationale for selecting a ≥4 mmHg cutoff by referencing prior studies and emphasizing the need for future work to align with standardized clinical criteria. This justification is included in the Discussion section. - Sample Homogeneity and Generalizability:
We have explicitly addressed the limitation related to the homogeneity of our cohort in the Conclusions. While this design controls for confounding factors, we agree that it limits external validity. We now emphasize the need for broader demographic inclusion in future research. - IOP Measurement Timing:
We agree that assessing IOP recovery over time would yield more comprehensive insights. We now recommend, in the IOP Measurement Protocol Section, that future studies include post-exercise measurements at intervals such as 15, 30, and 60 minutes. - Statistical Reporting:
In the Statistical Analysis section, we added a note about the lack of multiple comparison corrections and the absence of confidence intervals, and acknowledged that these are limitations. We will include confidence intervals in future analyses where appropriate. - Regression Analysis Clarification:
We clarified that the regression analyses are exploratory due to sample size limitations and added effect estimates. Specifically, we report that in weightlifters, maximum workload significantly predicted post-exercise IOP (R² = 0.32, p = 0.04). No such relationship was found in marathon runners. These findings are visualized in the new Figure 2 (scatterplot with regression line and 95% CI). - Figure Additions:
- Figure 1: Mean IOP values pre- and post-training for both athlete groups, with standard deviation error bars.
- Figure 2: Regression plot showing the relationship between maximum workload and post-exercise IOP in weightlifters.
- We have critically revised the relevant section to ensure that our interpretations remain strictly aligned with the data collected and avoid overextension into unmeasured mechanisms. Specifically, we have removed or rephrased speculative content related to cortisol dynamics and neuroprotective factors such as brain-derived neurotrophic factor (BDNF), which were mentioned in the original draft to contextualize findings within broader physiological literature. While these mechanisms are supported by prior research, we acknowledge that they were not directly investigated in our study. To clarify this point, we have added a paragraph to the end of the Discussion section (highlighted in red in the revised manuscript). This revision ensures that our conclusions remain data-driven while transparently acknowledging the potential—but untested—relevance of hormonal and neuroprotective mechanisms.
- We have revised the discussion to adopt a more moderate tone when referencing physiological mechanisms that may contribute to intraocular pressure (IOP) changes in weightlifters. Specifically, we have rephrased statements related to the Valsalva maneuver and episcleral venous pressure to present these mechanisms as possible contributors, rather than definitive explanations and we added primary literature references to support these physiological interpretations, which describe the relationship between intrathoracic pressure, episcleral venous outflow, and IOP elevation during exertion. We also integrated a new paragraph in the Discussion section (highlighted in red in the revised manuscript) to acknowledge the speculative nature of these pathways and ensure appropriate framing.
- We have carefully revised the manuscript to eliminate redundancies and improve overall conciseness and flow throughout the text.
- We have updated the reference list by replacing outdated sources with more recent and relevant literature to ensure the manuscript reflects current research.
- We appreciate the reviewer’s insightful suggestion. In response, we have added a dedicated paragraph to the Discussion section that clearly outlines the study’s limitations, including the small and homogeneous sample, limited generalizability, and lack of longitudinal follow-up. This paragraph also outlines directions for future research to address these constraints.
We hope these revisions sufficiently address the reviewer’s concerns and enhance the clarity, rigor, and scientific value of our manuscript. We are grateful for the opportunity to improve our work based on your feedback.

Reviewer 2 Report
Comments and Suggestions for Authors
The study is interesting and well done. Just a few points should be clarified, which include the following:
- Abstract, page 1, line 12: I don't know any other optic neuropathies secondary to elevate ocular pressure other than glaucoma. I suggest eliminating "and other optic neuropathies”
- Page 3, line 107: who is the principal investigator? Please add the initials.
- Page 3, line 119: I believe it is very difficult, even by dilating the pupil, to examine the peripheral retina using a direct ophthalmoscope.
- Page 3, line 122 and 123: add the name and location of the manufacturing company after “Goldmann applanation tonometer” and “IC200 rebount tonometer”
- Page 4, line 134: the Authors state that “all measurements took place in the hospital’s ophthalmology unit”. Then they state that “IOP was measured…within five minutes following the completion of the training session”. Is it really possible for athletes to reach the ophthalmology unit within five minutes of finishing an exercise session?
- Page 5, line 200: I suggest to remove “slightly lower than in the marathon runners” which is pleonastic.
- Discussion: although the activity of marathon runners is generally aerobic, in certain phases of the race and in middle-distance runners it often becomes anaerobic. It would be interesting to study the changes in IOP after this type of more demanding physical exercise. Authors could insert a short comment regarding this topic.
- References: check the abbreviations of some journals (Ref #6, 16, 23…)
Author Response
We sincerely thank the reviewer for their thoughtful and constructive comments, which have significantly improved the quality and clarity of our manuscript. We have carefully addressed all the suggestions as follows:
- Abstract (Page 1, Line 12): The phrase “and other optic neuropathies” has been removed, as suggested, to maintain accuracy and clarity.
- Page 3, Line 107: The initials of the principal investigator (F.M.) have been added for clarity.
- Page 3, Line 119: We acknowledge the reviewer’s observation. The text has been revised to reflect the limitations of direct ophthalmoscopy for peripheral retinal assessment.
- Page 3, Lines 122–123: The manufacturer name and location have been added after “Goldmann applanation tonometer” (Haag-Streit, Bern, Switzerland) and “IC200 rebound tonometer” (Icare Finland Oy, Helsinki, Finland).
- Page 4, Line 134: We clarified that pre- and post-training IOP measurements were conducted directly at the sports center, where a slit-lamp with a Goldmann tonometer was temporarily installed. The earlier phrasing has been revised for consistency.
- Page 5, Line 200: The phrase “slightly lower than in the marathon runners” has been removed to avoid redundancy.
- Discussion: In response to the valuable suggestion, we added a brief comment discussing the potential anaerobic component in certain phases of endurance events and its possible implications for IOP, with an appropriate citation (Billat et al., 2003).
- References: All journal abbreviations have been reviewed and corrected according to the appropriate standardized formats, including References #6, #16, and #23.
We hope these revisions meet the reviewer’s expectations and enhance the clarity and rigor of our manuscript. Thank you once again for your helpful feedback.
